# Children's views on research without prior consent in emergency situations: a UK qualitative study

Louise Roper,[1] Frances C Sherratt,[1] Bridget Young,[1] Paul McNamara,[2] Angus Dawson,[3] Richard Appleton,[4] Esther Crawley,[5] Lucy Frith,[1] Carrol Gamble,[6] Kerry Woolfall[1]

[1]Institute of Psychology Health and Society, University of Liverpool, Liverpool, UK
[2]Department of Child Health, Institute of Translational Medicine, University of Liverpool, Liverpool, UK
[3]School of Public Health, University of Sydney, Sydney, New South Wales, Australia
[4]Neurology Department, Alder Hey Children's Hospital, Liverpool, UK
[5]Bristol Medical School, University of Bristol, Bristol, UK
[6]Clinical Trials Research Centre (CTRC), University of Liverpool, Liverpool, UK

**Correspondence to**
Dr Kerry Woolfall;
K.Woolfall@liverpool.ac.uk

## ABSTRACT

**Objectives** We explored children's views on research without prior consent (RWPC) and sought to identify ways of involving children in research discussions.
**Design** Qualitative interview study.
**Setting** Participants were recruited through a UK children's hospital and online advertising.
**Participants** 16 children aged 7–15 years with a diagnosis of asthma (n=14) or anaphylaxis (n=2) with recent (<12 months) experience of emergency care.
**Results** Children were keen to be included in medical research and viewed RWPC as acceptable in emergency situations if trial interventions were judged safe. Children trusted that doctors would know about their trial participation and act in their best interests. All felt that children should be informed about the research following their recovery and involved in discussions with a clinician or their parent(s) about the use of data already collected as well as continued participation in the trial (if applicable). Participants suggested methods to inform children about their trial participation including an animation.
**Conclusions** Children supported, and were keen to be involved in, clinical trials in emergency situations. We present guidance and an animation that practitioners and parents might use to involve children in trial discussions following their recovery.

## INTRODUCTION

Informed consent for participation in research is a key principle of good clinical practice and protects an individual's right to autonomy.[1 2] Children under the age of 16 years are not legally permitted to give consent for their participation in clinical trials of drug treatments.[3] Nevertheless, the United Nations Convention on the Rights of the Child,[4] international legislation[5] and research guidelines[6] recommend that children should be involved in decision making processes in a way that is age appropriate to the child and the context of their family. The term assent is widely used and refers to when a child's wishes are taken into consideration in research decision making. It is recommended that assent

### Strengths and limitations of this study

► This is the first study to explore the views and acceptability of research without prior consent (RWPC) among children with experience of receiving emergency treatment for life-threatening conditions.
► We provide guidance and an animation to assist trial practitioners in involving children in RWPC discussions in collaboration with parents.
► Participants did not have personal experience of RWPC, so findings were limited to children's views on hypothetical scenarios.
► Participants lived in areas with varied levels of social deprivation. Despite attempts to further maximise sample diversity by use social media, the majority of children were recruited from one UK children's hospital.

is sought from children before they take part in research, where informed consent is not possible, although the role of assent has been criticised for being unclear in research discussions with children.[7 8]

Neither informed consent nor assent are appropriate or feasible in acute paediatric emergency situations,[9] as children are incapacitated and the time taken to discuss research with parents could delay lifesaving treatments.[10 11] To facilitate crucial clinical research and to advance evidence-based emergency medicine,[12] changes to international legislation have allowed doctors to enter patients into clinical research without prior consent (RWPC).[3 13–16] In the UK, RWPC, also known as deferred consent, involves approaching parents or legal representatives after the investigational treatment has been given and seeking permission for the use of data already collected as well as any continued participation in the trial.[17]

Guidance[18] and legislation[15] state that where possible, children should be involved in RWPC discussions once they have recovered. However, practitioners report that

RWPC discussions with parents take place at the bedside, shortly after the emergency situation has passed and that children are rarely involved at this point because they are too sick.[19] It is therefore likely that most children randomised into UK trials conducted in time critical situations have no knowledge or involvement in the decision about their participation in emergency research.

Despite growing literature on the views of parents and practitioners regarding paediatric trials conducted in life-threatening situations,[18 20 21] the opinions of children have not been investigated. This is an important omission as it is they who are most directly affected in this process. We aimed to explore children's perspectives on paediatric emergency research, including their views on the acceptability of RWPC and ways of involving them in RWPC discussions when they have recovered.

## METHODS

### Study design and setting

We chose a qualitative interview design to help us explore children's views in a flexible, child friendly way and to provide insight into children's perspectives on RWPC.[22 23] We recruited children for interviews through a children's hospital in the North West of England, supplemented by online-advertising to promote sample diversity.

### Patient and public involvement

Six members of the National Institute for Health Research (NIHR) Young Person's Advisory Group (YPAG) joined our 'Voices Advisory Group' to inform all aspects of the study. This included their collaboration in the development of recruitment materials, a topic guide, which included age-appropriate explanations of key topics (eg, RWPC; see online supplementary file 1) and vignette (see online supplementary file 2). KW and LR discussed initial findings with the Voices Advisory Group to assist analysis and interpretation. Members of the Voices Advisory Group also presented the findings with KW to a wide audience (at conferences and University events).

### Selection of participants

Children were eligible if they were aged 7–15 years; had written parental/legal representative consent to participate; had received emergency treatment in hospital in the previous 12 months; had capacity to assent and an acceptable standard of English.

Two clinical practitioners identified and approached parents and children who met the inclusion criteria, explained the aims of the study and provided age appropriate information sheets. Parents of the children who wished to participate completed a contact form, which was posted to LR. For online recruitment, FS contacted relevant support groups and asked them to place a study advertisement (online supplementary file 3) on their Facebook page, Twitter account or website. The advert included details of how children or parents could contact the study team to register interest in participation.

### Interview design and conduct

LR, a health psychologist arranged interviews either in the family's home, hospital or by telephone depending on participant preference. Before interviews, LR explained the study, referring to the information sheets and consent/assent forms provided. Consent and assent forms were completed by parents and children before the interview began.

We began interviews by showing children the Nuffield Council's animation[6] on a laptop or iPad to help set the scene and explain clinical research participation in a child-friendly way. No child had previous experience of RWPC, although they all had experienced at least one episode of receiving emergency care. We presented RWPC neutrally so that children could give their views freely. We also reassured children that there were no right or wrong answers.

LR conducted all interviews. Recruitment and interviews were discontinued when data saturation was reached, that is, the point where no new themes were discovered in the analysis.[24] Interviews were audio-recorded, transcribed verbatim and then anonymised. Participants were given a £10 shopping voucher and a certificate of participation as appreciation for their time.

### Data analysis

Our approach to data analysis was interpretive and iterative, referring back and forth between developing analysis and gathering new data for evidence of children's views on research in emergency situations, the acceptability of RWPC and ways of involving children in discussions about research in this context.[25 26] A key consideration during analysis was that of catalytic validity, whereby findings should be relevant to future research and practice.[27] NVivo software assisted data organisation. Several members of the research team (KW, LR, FS, BY) contributed to the analysis to ensure analytical rigor.[26 28] We present selected interview quotations (with pseudonyms) that illustrate research themes across a range of participants. Where quotes have been shortened for brevity or to remove identifiable information, omitted text is marked with '…' and explanatory text is in brackets.

## RESULTS

### Participants

Sixteen children (aged 7–15 (mean 10.2) years) were interviewed. At the children's request, parents were present for most of the interviews (n=12/16, 75%). Most participants had chronic asthma (n=14/16, 87%) and were recruited by a paediatric doctor (PM) in an outpatient clinic or hospital ward (n=2/16, 13%). Two participants suffered with anaphylaxis and were recruited via Facebook. Interviews averaged 34 min in length, ranging from 21 to 57 min (table 1). Participants' postcodes indicated that 9/16 (56%) participants lived in areas of high deprivation (Indices of Multi-Deprivation (IMD) decile 1–3), 5/16 (31%) lived in areas of moderate deprivation

**Table 1** Participant and interview details

| Pseudonym | Gender | Age | Health condition | Parent present |
|---|---|---|---|---|
| Chloe | Female | 12 | Asthma | Yes |
| Charlie | Male | 7 | Asthma | Yes |
| James | Male | 11 | Asthma | Yes |
| Emily | Female | 15 | Asthma | No |
| Niamh | Female | 15 | Asthma | Yes |
| Kaitlin | Female | 8 | Asthma | Yes |
| Joseph | Male | 7 | Asthma | Yes |
| Josh | Male | 11 | Asthma | Yes |
| Mia | Female | 9 | Asthma | Yes |
| Daisy | Female | 7 | Asthma | Yes |
| Kevin | Male | 8 | Asthma | Yes |
| Patrick | Male | 9 | Asthma | No |
| Lola | Female | 7 | Asthma | Yes |
| Ryan | Male | 8 | Asthma | Yes |
| Tom | Male | 15 | Anaphylaxis | No |
| Tilly | Male | 15 | Anaphylaxis | No |

(IMD decile 4–6), while 2/16 (13%) children lived in the least deprived (IMD decile 7–10) areas of the UK.[29]

### Support for RWPC, but only if a trial treatment was judged safe

All children voiced support for RWPC in emergency situations where a child's life was in danger, such as during an asthma attack or seizure. They explained that while parents and children should usually be asked for 'permission' before the research takes place, RWPC was acceptable in emergency research as it is important that doctors were able to give trial medicines without any delay in this situation.

> **Chloe, aged 12:** *most of the time (in research) you should ask the permission from the parents and the children… I think they can do that (research) without asking for permission because if it's an emergency and you have to give the medicine.*

The researcher (and the vignette of Daniel, see S3), informed children that only medicines that were believed to be safe would be used in paediatric trials. Children viewed this information as important, indeed, many stated that RWPC was only acceptable if doctors thought the treatment being tested was safe.

> **Josh, aged 11:** *If, say, like you're giving them a drug that hasn't been proven to have high probability of being safe, I don't think it would be good to give it them, but if it was probably going to be safe for them, I think they'd be fine.*

One child supported RWPC in the hope that the treatment being tested would help children to recover more quickly and to inform future paediatric medicine.

However, he warned that parents may have a different perspective and may be angry to find out that their child had been entered into a trial without informed consent if it turned out that the treatment was not effective.

> **Researcher:** *So what do you think about doing research without asking parents or children when they're very sick?*
>
> **James aged 11:** *I think the parents would be a bit ticked off like because they just want to get their child better, like I know my mum definitely would be…I would like it because if that drug did work on the off-chance then I would want to tell people (other children with chronic health conditions).*

### Misconceptions and misunderstandings

Many children appeared to understand that trials '*try to see if*' the medicine being investigated makes children '*better or not*' (Lola aged 7). However, a few children appeared to hold a misconception that research participation would have similar benefits to clinical care. For some children, this misconception seemed to be linked to the involvement of clinicians in the trial. Children stated they would not be upset about being entered into a RWPC trial as long as '*the doctors know*' (Emily aged 15) about their participation. As the quotes below suggest, children trusted clinicians to have their best interests at heart and wanted assurance that doctors and nurses would know about, and have approved, their involvement in a trial.

> **Researcher**: *So what if they gave you a new type of nebuliser and not told you? Do you think if it's that type of thing you should have been asked (about beforehand), or do you still think it's (RWPC is) okay?*
>
> **Emily aged 15**: *The doctors know (about being given a trial intervention)?*
>
> **Researcher:** *The doctors yes.*
>
> **Emily aged 15:** *Oh yes, definitely.*
>
> **Researcher:** *Would you be upset that they have tried something on you and not asked you first?*
>
> **Ryan aged 8:** *No.*
>
> **Researcher:** *Why is that?*
>
> **Ryan aged 8:** *Because the doctors and the nurse, they know.*

Others believed that within a trial, children would receive the 'right' treatment, while another implied that RWPC would only be acceptable if the intervention was effective:

> **Daisy aged 7:** *He'll (Daniel) get the right medicine to make him feel better.*
>
> **Tilly aged 14**: *I don't see the problem if it's going to stop it (anaphylaxis).*

### Why children want to be informed and involved in RWPC discussions

Children felt strongly they should be involved in RWPC discussions when they were feeling better. Several asked rhetorically, 'Why shouldn't we be told?' when asked about their participation in research. Irrespective of their

age, children reasoned they should be informed about the treatment received in a trial because '*it's going to impact on their body*' (Tilly, aged 14) not their parents' bodies. Indeed. Tilly added that not being informed about their participation in a trial would '*feel like I'd been sort of left out, like my opinions and views sort of, I don't get to express them*' suggesting that not being included in RWPC discussion would deny children their voice and the opportunity to have a say in their healthcare treatment.

A few children described wanting to be informed about their involvement in research and the trial results to '*know what the trial had proved*' (Chloe aged 12) for their own chronic health condition. They expressed a wish to help inform future knowledge about treatments for other children by finding '*out about new medicines on the computer that are better than other ones and then they could give us it and then it might make us better*'. (Joseph aged 7).

Finally, two young boys (<12 years) wanted to know about their participation because they viewed medical research as exciting. For them, children should be informed about their participation in research because '*it could be fun to take part in stuff*' (Joseph, aged 7) and '*it's quite cool to be part of like a scientific medical thing*' (Josh, aged 11).

### Who should explain RWPC to children?

Most children commented that the best person to explain that they had participated in a study while they were ill would be a doctor or nurse as they would be the most knowledgeable.

> **Tilly, aged 14**: *if it's a nurse who knows all about it and then also has studied kids, they can then help with that as well, help them do it, understand what's happened and go through it. But I don't think I'd listen if my mum told me. I would want it more, the medical advice really.*

> **Kaitlin, aged 8**: *By the doctor … because he can explain it more.*

Some envisaged they would have questions about the research which they would like a doctor or nurse to answer, such as: '*How could it affect me? What would be like the aftermath of it?*' (James aged 11), '*What could the risks be?*' (Joseph aged 7) as the doctor or nurse would be best placed to answer.

Although some children said they '*wouldn't mind who told*' (Chloe aged 12) them about their participation in a trial, a few younger children stated that they would prefer to be informed by their mothers because it may '*feel a bit awkward*' talking to the doctors and its '*easier with my mum*' (Mia aged 9).

### Children's role in decisions about the use of their information

Children varied in their views about the role of children in RWPC decisions when they were feeling better. Younger children were typically happy for their parents (mainly mothers) to make the final decision about the use of their information in a trial.

> **Kevin aged 8**: *Up to my mum.*

> **Joseph, aged 7**: *I would listen to my mum and the doctors because older people can sometimes be cleverer than younger people than them, because they've had more years to learn about stuff.*

Participants of all ages echoed this by commenting that children younger than themselves, '*don't really understand […] what the research is about*' (Chloe aged 12) and '*might be a bit too young*' (Joseph aged 7) to make such decisions and that for such children it should be a parent's role to make a decision about the use of their information for research purposes.

Similarly, some children stated that they would accept their parents' decision, even if it was at odds with their own views, while others wanted to share in decision-making with their mothers.

> **Emily aged 15**: *My little brother, he's 11 and I think he's sensible so he'd probably choose the right thing. But if you're a bit younger I think it should be up to your mum.*

> **James, aged 11**: *If mum said no and I said yes, I would just listen to mum.*

> **Researcher**: *What about if you wanted to be in it but mum said no, then who should (the doctor) listen to?*

> **Chloe, aged 8**: *Both of us.*

In contrast, a few teenage participants felt that '*they should be able to actually make a medical choice for themselves because it's impacting them in the end*' (Tilly aged 14). Therefore, their decision about the use of their data and involvement in any follow-up, should override, or at least have greater value, than their parents' decisions.

> **Researcher**: *What if your mum says yes, that's fine and you say no? Then who should be in charge?*

> **Emily aged 15**: *Me.*

### Child friendly resources to help communicate RWPC: introducing the 'You took part in research animation'

Finally, we sought participants' suggestions on the most appropriate resources to help to engage with children in RWPC discussions when they were feeling better. Participants emphasised the need for a face to face discussion so doctors or nurses could '*explain it to you, or give you a leaflet, then…you can ask questions in person*' (Chloe aged 12). While a few suggested that leaflets, websites and picture books would be useful in explaining RWPC, most favoured an online animation that could be used either in hospital as part of a face to face discussion, or '*at some point when I was at home*' (Tom aged 13) to '*make sure they understand everything properly*' (Josh aged 11) to explain RWPC.

> **Patrick aged 9**: *because some doctors don't know how far to explain stuff when they are particularly young kids. So even the doctors to have on their phone or their iPad and go this is what we've done, and then they watch it and then they explain a bit more then.*

The findings of this study encouraged the study team to develop an animation with the help of the Voices Advisory Group and three of interviewees in the current study who wished to help with this. The 'You took part in research' animation available to view on YouTube (https://www.youtube.com/watch?v=_Fs1yUxeBFQ), focuses on explaining why consent could not be sought before a child took part in research.

## DISCUSSION

We believe this is the first study to explore the views and acceptability of RWPC among children with experience of receiving emergency treatment in hospital. The children we interviewed supported RWPC in the hope that such research would contribute to developments in children's emergency medicine for the benefit of themselves and other children. However, RWPC was only acceptable to children where trial interventions were believed by the clinicians caring for them to be safe. This finding also echoes previous literature on therapeutic misconceptions about trial participation among adults,[30] as some children held the misconception that research participation would have similar benefits to clinical care. Others appeared to trust that doctors would know about their trial participation and act in their best interests. Others believed that within a trial, children would receive the 'right', treatment. Such findings suggest that trial information materials and practitioner RWPC discussions should clearly explain that doctors don't know which intervention is the best, which is why a trial is needed. This information should be supplemented with details of other treatments given (if applicable) and any additional monitoring that occurred.

This study has strengths and limitations. This is the first study to explore the views and acceptability of RWPC among children with experience of receiving emergency treatment in hospital. We strengthened our qualitative sampling by conducting interviews until no new relevant knowledge was obtained from new participants (data saturation). All participants had a chronic health condition and experienced multiple hospital admissions for emergency lifesaving treatments. However, our findings do not reflect the views of acutely ill children admitted to hospital for the first time. Participants did not have personal experience of emergency medicine research so findings were limited to children's views on hypothetical scenarios about RWPC. Finally, the study findings are mainly limited to the views of children recruited from an asthma clinic in one UK children's hospital. Nevertheless, there was variation in the sample, as children lived in areas with differing levels of social deprivation. Despite the successful use of websites and social media advertising to recruit parents to similar qualitative studies,[21 31] only two children responded to social media advertisement. The low response to this indicates social media such as Facebook and Twitter may not be an effective route for recruiting children to research. Placing adverts on social media more commonly used by children (eg, Snapchat)[32] may have been more successful.

Children's support of RWPC in emergency situation reflects the findings of previous studies on this topic involving trial practitioners and parents.[18 19 21 31 33] Where studies[18 19 31] have shown that parents can have initial negative responses to RWPC, children did not appear to be shocked or surprised at this alternative to giving informed consent before the research intervention. This may reflect their lack of familiarity with standard research consent processes, or indeed, their lack of previous involvement in decisions about their healthcare. Our findings suggest that children trust doctors and parents to make appropriate decisions about their participation in research when they are unable to do so themselves. However, children were clear that when they were feeling better, adults (particularly mothers and doctors) should inform them of their trial participation and discuss the use of their data.

Children in our study acknowledged the important role that children, parents and doctors played in research decision making. This reflects guidelines recommending that decisions about research should involve discussion and information exchange between children, their parents and doctors.[6 12 34–36] Some described the prospect of research participation as exciting and fun, while others seemed to want the satisfaction of knowing if they had contributed to research that might improve future treatments for both themselves and others. Children wanted to be involved in decisions regarding their own healthcare.[37 38] Younger children were happy for their parents to take the lead or a shared role in research decision-making. A few teenagers felt their views should carry greater weight than their parents suggesting that practitioners should be aware of the possibility that teenagers may wish to take a lead in such decisions. As previous literature[19 21 31] and Voices study findings all show support for RWPC, we do not anticipate divergence in child and parent views on consent in this situation, mainly because the intervention has already been given and children have commonly recovered when research discussions take place. The challenge is ensuring children are given the opportunity to be (retrospectively) informed about their research participation and actively involved in research discussions and decision making with their parents.

We acknowledge there may be practical barriers to involving children in RWPC discussions, as they may not be well enough or have the capacity to be involved before discharge from hospital. Nevertheless, our findings[19] indicate that when they have recovered, children are not given the opportunity to know they have participated in research or a say in whether their data continue to be used in research. Based on the accounts of the children we interviewed, this is unlikely to be acceptable to children who participate in trials to evaluate interventions used in emergency care. As one child we interviewed eloquently stated, *'it's not impacting their life, it's impacting your own'* (Tilly, aged 15).

## Box 1  Guidance to inform and involve children in RWPC discussions

Children should be given an opportunity to be involved in making decisions about the use of their data in the trial and their continued enrolment if they have capacity.

► When children cannot be involved in trial consent discussions due to a lack of capacity, provide age appropriate trial information (such as a leaflet or website) to parents for children to read when they have recovered.

► Include a link to the 'You took part in research animation' https://www.youtube.com/watch?v=_Fs1yUxeBFQ) to help explain why consent could not be sought before a child took part in research.

► Provide research team contact details so that parents or children can discuss any aspect of the trial at a later date.

Box 1 (guidance to inform and involve children in RWPC discussions) provides an update to CONNECT study RWPC guidance,[39] which aims to assist trial practitioners in involving children in RWPC discussions in collaboration with parents. To respect the wishes of children we encourage trial staff to involve them in trial discussions in hospital if they are well enough. The 'You took part in research' animation can be used to support these discussions. If this is not possible before children are discharged from hospital. Age-appropriate information can be provided to facilitate family discussions about trial participation at home, including contact details if children wish to discuss their participation with the trial team. While our focus in this paper has been children (under 16 years), we believe our findings also relate to trial discussions with young people (16–18 years).

Further research is needed to explore the experiences of children with first-hand experience of RWPC and associated discussions.

**Acknowledgements** We would like to thank our Young Person's Advisory Group (YPAG) for their valuable insights into the study design, conduct and interpretations of results as well as the children who took part in the study and helped develop the animation. We would also like to thank Beth Morris and Paige Karadag for their comments on the final draft.

**Contributors** KW conceived the study. KW, BY, LR and FCS designed the study. PM, LR and FCS recruited children to the study. LR conducted the interviews. LR, FCS and KW analysed the data. KW and LR drafted the paper. KW, LR, FCS, BY, PM, AD, RA, EC, LF and CG contributed to revisions and approved the final draft. KW takes responsibility for the paper as a whole.

**Funding** The study was funded by the MRC Hubs for Trials Methodology Research (project R42) and supported by Wellcome Trust ISSF award.

**Disclaimer** The lead author affirms that the manuscript is an honest, accurate and transparent account of the study being reported; that no important aspects of the study have been omitted and that any discrepancy from the study as originally planned has been explained. The report is independent research supported by the National Institute for Health research (Senior Research Fellowship, Prof Esther Crawley, SRF-2013-06-013). The views expressed in this publication are those of the authors and not necessarily those of the NHS, the National institute for Health Research or the Department of Health.

**Competing interests** None declared.

**Patient consent** Not required.

**Ethics approval** A National Health Service Research Ethics Committee (15/NW/0915) provided ethical approval.

**Provenance and peer review** Not commissioned; externally peer reviewed.

**Data sharing statement** No additional data are available.

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
