## [Reviewer comments · BMJ Open]

ARTICLE DETAILS

TITLE (PROVISIONAL)	CHILDREN'S VIEWS ON RESEARCH WITHOUT PRIOR CONSENT IN EMERGENCY SITUATIONS: A UK QUALITATIVE STUDY
AUTHORS	Roper, Louise; Sherratt, Frances C.; Young, Bridget; McNamara, Paul; Dawson, Angus; Appleton, Richard; Crawley, Esther; Frith, Lucy; Gamble, Carrol; Woolfall, Kerry

VERSION 1 – REVIEW

REVIEWER	Kristien Hens Universiteit Antwerpen, Belgium
REVIEW RETURNED	17-Apr-2018

GENERAL COMMENTS	This is a good paper based on qualitative research with children who have experience with emergency care, on their views on participating in research without prior consent. As such, it fills a gap in knowledge, as the opinions of children have been traditionally largely ignored. I had no real comments on methodology, results or discussion. Maybe a formal definition of assent could be given in the introduction for those not familiar with this terminology.
--

REVIEWER	Karen Goddard BC Cancer, Vancouver Center Canada
REVIEW RETURNED	23-Apr-2018

GENERAL COMMENTS	Interesting and unique work - it would be worthwhile exploring this question further in adolescent patients alone
---

VERSION 1 – AUTHOR RESPONSE

Reviewer 1

This is a good paper based on qualitative research with children who have experience with emergency care, on their views on participating in research without prior consent. As such, it fills a gap in knowledge, as the opinions of children have been traditionally largely ignored. I had no real comments on methodology, results or discussion. Maybe a formal definition of assent could be given in the introduction for those not familiar with this terminology.

Response: We would like to thank the reviewer for their comments. We have added a definition of assent to the background section.

Reviewer 2

Interesting and unique work - it would be worthwhile exploring this question further in adolescent patients alone

Response: We would like to thank reviewer 2 for their suggestion on exploring this question further with adolescent patients and can confirm we aim to explore this in the future.

Editorial office comments:

You have cited reference #29 right after reference #26 and reference number #34 right after 31 which makes your citations incorrect. Please review again your main document and ensure that all references will be cited and will appear in ascending order.

Response: We have refreshed endnote and can confirm that the reference numbers are now in ascending order unless the reference has been used in an earlier part of the document.

- Please provide a more detailed contributor ship statement. It needs to mention all the names/initials of authors along with their specific contribution/participation for the article.

- Response: this additional detail has been added

We have also addressed the other submission queries raised.